# 24-Epibrassinolide Promotes Fatty Acid Accumulation and the Expression of Related Genes in *Styrax tonkinensis* Seeds

**DOI:** 10.3390/ijms23168897

**Published:** 2022-08-10

**Authors:** Chen Chen, Hong Chen, Chao Han, Zemao Liu, Fangyuan Yu, Qikui Wu

**Affiliations:** 1Collaborative Innovation Centre of Sustainable Forestry in Southern China, College of Forest Science, Nanjing Forestry University, 159 Longpan Road, Nanjing 210037, China; 2State Forestry and Grassland Administration Key Laboratory of Silviculture in Downstream Areas of the Yellow River, College of Forestry, Shandong Agricultural University, Tai’an 271018, China

**Keywords:** 24-epibrassinolide, fatty acid biosynthesis, GC-MS, *Styrax tonkinensis*, transcriptome

## Abstract

*Styrax tonkinensis*, whose seeds are rich in unsaturated fatty acids (UFAs), is a high oil value tree species, and the seed oil has perfect biodiesel properties. Therefore, the elucidation of the effect of 24-epibrassinolide (EBL) on fatty acid (FA) concentration and the expression of FA biosynthesis-related genes is critical for deeply studying the seed oil in *S. tonkinensis*. In this study, we aimed to investigate the changing trend of FA concentration and composition and identify candidate genes involved in FA biosynthesis under EBL treatment using transcriptome sequencing and GC-MS. The results showed that 5 μmol/L of EBL (EBL5) boosted the accumulation of FA and had the hugest effect on FA concentration at 70 days after flowering (DAF). A total of 20 FAs were identified; among them, palmitic acid, oleic acid, linoleic acid, and linolenic acid were the main components. In total, 117,904 unigenes were detected, and the average length was 1120 bp. Among them, 1205 unigenes were assigned to ‘lipid translations and metabolism’ in COG categories, while 290 unigenes were assigned to ‘biosynthesis of unsaturated fatty acid’ in KEGG categories. Twelve important genes related to FA biosynthesis were identified, and their expression levels were confirmed by quantitative real-time PCR. *KAR*, *KASIII*, and *accA*, encoding FA biosynthesis-related enzymes, all expressed the highest at 70 DAF, which was coincident with a rapid rise in FA concentration during seed development. *FAD2* and *FATB* conduced to UFA and saturated fatty acids (SFA) accumulation, respectively. EBL5 induced the expression of FA biosynthesis-related genes. The concentration of FA was increased after EBL5 application, and EBL5 also enhanced the enzyme activity by promoting the expression of genes related to FA biosynthesis. Our research could provide a reference for understanding the FA biosynthesis of *S. tonkinensis* seeds at physiological and molecular levels.

## 1. Introduction

The consumption of biological oil has increased at an annual rate of 5% over the past 50 years due to the growing global population and rapid industrialization. Biodiesel is a renewable, safe, and green energy, which is regarded as an alternative resource to alleviate the energy crisis [1,2]. Animal fats are low in unsaturated fatty acids (UFAs) and contain many impurities [3]. On the contrary, plant oils are abundant in UFAs and easy to be extracted [4]. Hence, high oil content tree species receive much more attention. Fatty acids (FAs) are the main components of lipids, holding practical values in chemical and pharmaceutical industries, and could be utilized as crucial raw materials for food processing [5,6]. Many UFAs can regulate blood lipids, enhance immunity, prevent cardiovascular diseases, and so on [7]. Therefore, improving the content and quality of FAs is meaningful to daily life and production practice, and will become one of the momentous objectives of oil tree breeding.

FA biosynthesis is mainly carried out in plastids and controlled by the complex transcriptional and enzymatic regulatory network. For example, acetyl-CoA carboxylase (ACCase) catalyzes the carboxylation of acetyl CoA to form malonyl CoA, which is a pivotal regulatory step for FA biosynthesis and lipid formation, while FA desaturases (FADs) are the important enzymes that bring double bonds in FA acyl chains in a stepwise manner starting from stearic acid [8,9]. It was demonstrated that the expression of diacylglycerol acyltransferase (*DGAT*) affected seed oil content and FA composition [10]. The overexpression of *AtDGAT1* caused the DGAT activity in transgenic *Arabidopsis thaliana* seeds to become 10% to 70% higher than that of the wild type, and the oil content is also higher in transgenic plants [11]. Thus, studies on the biosynthesis and accumulation of FAs associated with different enzyme genes are of great significance for understanding the interaction between multiple genes and further illuminating their genetic regulatory networks [12].

*Styrax tonkinensis*, a member of the family Styracaceae, is native to Laos and Vietnam and is widely distributed in the southern provinces of China [13]. The flowers of *S. tonkinensis* are in strings with a light fragrance and can be used as medicine to relieve pain [14]. Its bark is the source of benzoin, which can be used as a flavoring agent and can produce incense, perfume, and medicine [15,16]. Previous research has indicated that the FA concentration in *S. tonkinensis* seeds is high, and the UFA concentration increases with seed development, while the saturated fatty acid (SFA) concentration decreases [17,18]. At the same time, the molecular mechanism of FA biosynthesis in *S. tonkinensis* seeds has been elucidated [18]. Although *S. tonkinensis* possesses great potential as a woody biodiesel species with high oil content, its FA concentration and composition still need to be further improved to realize its industrialization. Breeding and cultivation are the main means to achieve this ascension; nevertheless, breeding is time-consuming. In terms of cultivation techniques, the exogenous spraying of 24-epibrassinolide (EBL) has been verified to contribute to FA accumulation in other tree species [19,20], and this promoting effect was realized by enhancing the activities of enzymes related to FA synthesis, increasing CO_2_ assimilation through improving stomatal conductance, regulating cellular lipid metabolism and so on [21,22]. However, its influence on FA accumulation in *S. tonkinensis* seeds and the underlying mechanism for FA biosynthesis remain unknown.

In higher plants, FA biosynthesis is normally coupled with seed development. Consequently, it is imperative to explore various FA biosynthesis-related enzyme genes at different developmental stages. The fruits treated with EBL on different days after flowering (DAF) were used as experimental materials in the present study and analyzed via transcriptome and GC-MS. Additionally, key genes for the biosynthesis of FA and their expression pattern under EBL treatment were identified through quantitative real-time PCR (qRT-PCR). The purpose of this study is to: (1) probe the impact of EBL on the accumulation of FA and alteration in composition in *S. tonkinensis* seeds; (2) assess the transcriptional profiles at 50, 70, 100, and 130 DAF; and (3) ascertain the functional genes encoding enzymes associated with FA biosynthesis.

## 2. Results

### 2.1. Effects of EBL5 on Fatty Acid Composition, Fatty Acid Concentration, and Fatty Acid Biosynthesis-Related Enzyme Activity

In the present study, we evaluated the dynamic patterns of FA concentration under EBL5 treatment during seed development, and the results were displayed in Figure 1a. The FA concentration increased drastically from 50 DAF to 70 DAF, and the concentration of FA in EBL5 (47.9%) was significantly higher than that in control (CK, 40.5%). Thereafter, the FA concentration followed a down-up trend with the second peak at 130 DAF. In this period, the discrepancy in seed FA concentration from EBL5-treated and untreated trees narrowed, and the difference between them was not significant. In general, the FA concentration in seeds from EBL5-treated trees was higher than that of untreated trees. Additionally, the FA content per seed increased continuously, and EBL5 greatly increased the FA content (Figure 1b). From 50 DAF to 70 DAF, the increase rate was large, then the rate became smaller. At 130 DAF, the FA content per seed in EBL5 could reach up to 48.13 mg, 33% higher than that in CK.

The compositions of FA under EBL5 and CK were determined, and the results were shown in Appendix A. There were 20 FAs in *S. tonkinensis* seeds, among which palmitic acid (C16:0), oleic acid (C18:1), linoleic acid (C18:2), and linolenic acid (C18:3) were major FAs. Their concentrations at 50 DAF in CK were 25.53%, 13.80%, 24.86%, and 18.63%, respectively. Afterwards, the concentration of C16:0 declined continuously, and the lowest concentration occurred at 130 DAF. The concentration of C18:1 and C18:3 descended first and then ascended, however, the concentration of C18:2 showed the opposite trend. EBL5 slightly increased the C18:1 and C18:3 concentrations at 70 DAF and 100 DAF but decreased the C18:2 concentration in the same period. These four major FAs accounted for 80.84–95.38% in seeds. Except for the compositions mentioned above, the concentrations of the rest of the FAs were relatively low, and stearic acid (C18:0) dominated among them. All FAs were classified into UFAs and SFAs, and the ratio of UFAs and SFAs was calculated and exhibited in Figure 1c. It could be found that the ratio rose from 50 DAF to 130 DAF, and the values were all greater than 1, indicating that UFAs concentration was always higher than that of SFAs. Abundant UFAs were detected in the middle and late stages of seed development.

Generally, Figure 1d showed an ‘A’ trend of the activity of fatty acid synthase (FAS) in seeds from EBL5-treated and untreated trees, and the highest activity presented at 70 DAF. The minimum values of FAS activity in seeds from CK and EBL5-treated trees appeared at 100 DAF and 50 DAF, respectively, which were 33.23 nmol/min/mg protein and 57.88 nmol/min/mg protein. Although FAS activity under CK and EBL5 reached their maximum at 70 DAF, the biggest difference between them occurred at 100 DAF; that is, FAS activity under EBL5 was 4.46 times that of CK. During the seed development, the FAS activities in seeds from EBL5-treated trees were all higher than that of CK.

### 2.2. Transcriptome Sequencing Analysis

The seed samples under different treatments at four time points were analyzed using transcriptome sequencing. More than 5.96 GB of raw data were generated from each sample. In total, 616,586,008 and 584,476,454 clean reads were obtained from CK and EBL5, respectively, after removing the adaptor and low-quality reads, and 24 libraries were constructed (Table 1). All samples were assembled using Trinity, and Table 2 showed that a total of 117,904 unigenes were detected, and the average length was 1120 bp. The largest and smallest length of these unigenes were 16,018 bp and 201 bp, respectively. The average N50 length was 1705 bp with a GC content of 42.21%. It was clear that 39,371 unigenes (33%) were 200–500 bp in length, 36,701 (31%) were 501–1000 bp, 2858 (2%) were over 4500 bp, and the rest of the unigenes were distributed in 1001–4500 bp (Figure 2).

### 2.3. Functional Annotation and Classification

Basic Local Alignment Search Tool (BLASTX) was used to annotate all the assembled unigenes against six publicly available protein databases. A total of 66,457 unigenes were annotated in a database, among which more unigenes were annotated in Non-Redundant protein (NR), Clusters of Orthologous Groups (COG), and Gene Ontology (GO) (61,317, 55,114, and 50,708, respectively), the number of unigenes annotated in the Protein Family (Pfam, 48,409), and Swiss-Prot Protein (Swiss-Prot, 46,471) were close, and only 30,977 unigenes were annotated in Kyoto Encyclopedia of Genes and Genomes (KEGG) (Figure 3).

In this study, a total of 50,708 unigenes were involved in biological processes, cellular components, and molecular functions through the GO database (Figure 4). The biological process contained 23 sub-categories, among which ‘cellular process’ and ‘metabolic process’ possessed the most abundant unigenes (23,648 and 20,942, respectively). The cellular component included 14 sub-categories; of these sub-categories, ‘cell part’ (24,167), ‘membrane part’ (18,787), as well as ‘organelle’ (14,227) were more prominent. The molecular function was assigned into 17 sub-categories with the majority of the unigenes in ‘binding’ (29,083) and ‘catalytic activity’ (27,371).

A total of 55,114 unigenes were annotated in 23 COG functional categories, and the main category was ‘function unknown’ (28,769), suggesting that the function of more unigenes was worth further exploring (Figure 5). In addition, ‘posttranslational modification, protein turnover, chaperones’ and ‘translation, ribosomal structure and biogenesis’ were the other two major categories, which had 4232 and 4199 unigenes, respectively. Furthermore, 1205 unigenes were assigned to ‘lipid translations and metabolism’, indicating candidate genes that may be associated with *S. tonkinensis* seed oil accumulation.

A total of 30,977 unigenes were assigned to six KEGG categories, 20 sub-categories, and 524 KEGG pathways (Figure 6). Of these unigenes, most were found in ‘biosynthesis of unsaturated fatty acid’ (290), followed by ‘glycerophospholipid metabolism’ (116) and ‘glycerolipid metabolism’ (69). Moreover, the other lipid metabolic canonical pathways, such as ‘sphingolipid metabolism’, ‘fatty acid biosynthesis’, ‘arachidonic acid metabolism’, ‘fatty acid degradation’, ‘fatty acid elongation’, and ‘linoleic acid metabolism’, were also annotated in the current study.

### 2.4. Gene Expression Analysis

To reveal the dynamic expression patterns of the unigenes, the transcriptome profiles under different treatments from different time points were compared. A total of 1119 unigenes were differentially expressed in CK seeds between 50 DAF and 70 DAF, among which 702 unigenes were upregulated and 417 unigenes were downregulated (Appendix A, Figure 7). More unigenes (2540) had different expression patterns between 50 DAF and 100 DAF, with 811 upregulated unigenes and 1729 downregulated unigenes. Between 50 DAF and 130 DAF, 9389 unigenes were expressed differentially (5494 upregulated and 3895 downregulated). A total of 1790 unigenes had different expression patterns between 70 DAF and 100 DAF, including 378 upregulated unigenes and 1412 downregulated unigenes. Compared to 70 DAF, 8771 unigenes were found at 130 DAF, with 5196 upregulated unigenes and 3575 downregulated unigenes. Between 100 DAF and 130 DAF, 5360 unigenes were expressed differentially, including 3698 upregulated unigenes and 1662 downregulated unigenes.

EBL5 was applied on *S. tonkinensis* fruits to clarify their influence on the expression patterns of unigenes at different sampling times. The differentially expressed genes (DEGs) in seeds from EBL5-treated trees were much higher when compared to CK at 100 DAF and 130 DAF, nonetheless, it was the opposite at 50 DAF and 70 DAF (Appendix A, Figure 7). In the early stage, the highest DEGs number under EBL5 treatment was 328 (with 235 upregulated unigenes and 93 downregulated unigenes), which was found at 50 DAF. Hereafter, the DEGs number went up greatly, reaching the maximum at 100 DAF under EBL5 treatment (2249). We also observed that the number of upregulated genes was higher than that of downregulated genes from 50 DAF to 100 DAF under EBL5, but the opposite trend was noted at 130 DAF.

Here, we performed a Venn analysis to obtain the co-expression genes in all samples and the specific expression genes in each sample. As described in Figure 8, the number of co-expression genes among all samples was 21,595. The number of specific expression genes in CK showed an ‘upward-downward-upward’ trend during seed development, with the lowest number at 100 DAF (1033) and the highest number at 130 DAF (5510). As for the number of specific expression genes in seeds from EBL5-treated trees, the minimum and the maximum were found at 50 DAF and 130 DAF, respectively. Except at 70 DAF, EBL5 contained more specific expression genes than that in CK in the same period. Overall, the number of unigenes within each treatment fluctuated moderately from 50 DAF to 100 DAF, while the dynamic became drastic at 130 DAF.

### 2.5. Enrichment Analysis of DEGs

We subjected DEGs to the KEGG pathway to screen genes associated with FAs in *S. tonkinensis* seeds. The top 15 enriched pathways were ribosome, oxidative phosphorylation, protein processing in endoplasmic reticulum, glycolysis/gluconeogenesis, endocytosis, spliceosome, RNA transport, plant-pathogen interaction, pyruvate metabolism, glyoxylate and dicarboxylate metabolism, starch and sucrose metabolism, carbon fixation in photosynthetic organisms, peroxisome, fatty acid degradation, and phagosome. Additionally, the biosynthesis of unsaturated acids and fatty acid biosynthesis were also enriched.

### 2.6. Expression Levels of Genes Involved in Fatty Acid Biosynthesis

During the development of *S. tonkinensis* seeds, the expression levels of various genes at 70, 100, and 130 DAF were calculated by referring to the expression levels of genes at 50 DAF. The relative expression levels of 12 genes related to FAs biosynthesis were displayed in Figure 9. Among these genes, the relative expression of 3-ketoacyl-ACP reductase (*KAR*), stearoyl-ACP desaturase 2 (*SAD2*), acyl-CoA synthetase family member 3 (*ACSF3*), 3-ketoacyl-ACP synthase III (*KAS III*), 3-oxoacyl-ACP synthase of mitochondria (*OXSM*), and ACC carboxyltransferase (*accA*) shared the similar dynamic pattern with the concentration of FA from 50 DAF to 130 DAF, which was maximum at 70 DAF; simultaneously, EBL5 resulted in higher gene expression. As a result, these six genes were salient for FA biosynthesis in *S. tonkinensis* seeds. The relative expression of 3-ketoacyl-ACP synthase II (*KAS II*), enoyl-ACP reductase (*EAR*), acyl-ACP thioesterase B (*FATB*), and ACC biotin carboxyl (*accB*) decreased continuously in general, and EBL5 enhanced their expression. Inversely, the expression of long-chain acyl-CoA synthetases (*ACSL*) and fatty acid desaturase 2 (*FAD2*) showed an irregular trend, and CK possessed a relatively high expression. Ten genes were chosen for qRT-PCR validation, and the data were basically in accordance with RNA sequencing, indicating that the sequencing results were reliable and could be used for further analysis (Appendix A).

The FA biosynthesis was mainly carried out in plastids, and the precursor material for the biosynthesis was acetyl CoA [23]. Firstly, ACCase catalyzed the formation of malonyl CoA from acetyl CoA (Figure 10). Then, the FAs used malonyl CoA as a substrate to conduct continuous polymerization reactions to catalyze the extension of carbon chains and synthesize acyl carbon chains at the frequency of increasing two carbon atoms per cycle until the synthesis of SFAs of 16–18 carbon. In this process, 16:0-ACP and 18:0-ACP formed SFAs, such as C16:0 and C18:0, under the action of enzymes encoded by *FATB*. The enzymes encoded by *SAD* could promote the transformation of 18:0-ACP into 18:1-ACP, which in turn formed C18:1 under the action of *FATA*. EBL5 enhanced the expression of *KAR* and *KASIII*, which was conducive to carbon chain extension. At the same time, the expression of *FATB* in the seeds of *S. tonkinensis* treated with EBL5 increased, indicating that the exogenous spraying of EBL5 could promote the process of 16:0-ACP releasing C16:0, thus improving the accumulation of SFAs.

## 3. Discussion

### 3.1. Fatty Acid Accumulation and Compositions in S. tonkinensis Seeds

Biodiesel is a sustainable energy and provides an approach to resolve the energy crisis. Plant FAs are not only salient nutrients for human beings but also provide raw materials for industry [4]. Many plant seeds are rich in FAs and have the potential to produce biodiesel, such as lemon and *Xanthoceras sorbifolia* [24,25]. The seeds of *S. tonkinensis* are abundant in FAs, mainly C16:0, C18:1, C18:2, and C18:3, which represent the high-quality raw material for biodiesel production. During different sampling periods after flowering, FA concentration in seeds from untreated trees soared firstly, then dropped and increased at last, with the peak emerging at 70 DAF (40.5% DW, Figure 1a), which was identical to the result of [18]. In this study, we found that EBL5 increased the concentration of FAs in each period, but the extent was different. Meanwhile, the FA content in single seeds ascended dramatically from 50 DAF to 70 DAF, which was consistent with the trend of FA concentration and FA biosynthesis-related enzyme activities. Thereafter, the increase in FA content in single seeds slowed down. During seed development, EBL5 consistently enhanced the FA content in individual seeds. EBL treatment also favored the elevation of FA concentration in *Carthamus tinctoriu* seeds [26]. Exogenous treatment could increase the activity of enzymes related to FA biosynthesis, resulting in a higher concentration of FA [27]. The FAS activity in the seeds from EBL5-treated trees was always higher than that in the CK seeds during the whole sampling period, which provided strong evidence for the above conclusion.

In addition to quantity, the proportion of FAs is critically important for feedstock quality [28,29]. Twenty FA compositions were identified during seed development under different treatments, and these compositions were relatively steady. The main SFA was C16:0 with a downward trend from 50 DAF to 130 DAF (Appendix A), and EBL5 affected C16:0 concentration variously. For edible oils, UFAs play a key role in human health. Thus, increasing the concentration of UFAs is crucial for producing edible oil. As shown in Appendix A, 18C FAs were the main compositions of FAs in *S. tonkinensis* seed oil, and C18:1 and C18:2 reached the maximum at 130 DAF and 70 DAF, respectively. Hence, these data provided a reference for the accurate extraction of UFAs, which would help augment the value of FAs in *S. tonkinensis* seeds. Overall, EBL5 increased the concentration of C18:1 and C18:3 at 70 DAF and 100 DAF, while it increased the concentration of C18:2 at 50 DAF and 130 DAF. Moreover, the concentration of UFAs was always higher than SFAs during the development of *S. tonkinensis* seeds. The UFA/SFA ratio increased from 1.55 at 50 DAF to 5.44 at 130 DAF in CK seeds (Figure 1c). The maximum UFA/SFA under EBL5 also occurred at 130 DAF.

### 3.2. Expression Levels of the Genes Involved in Fatty Acid Biosynthesis

FA de novo biosynthesis began with the conversion of acetyl-CoA to malonyl-CoA, which was catalyzed by ACCase. Then, malonyl-CoA:ACP S-malonyltransferase (MAT), the principal substrate for the subsequent elongation, transferred malonyl-CoA to the malonyl group. Next, KAR, EAR, 3-hydroxyacyl-ACP dehydratase (HAD), and 3-ketoacyl-ACP synthase I (KASI) resulted in carbon chain extension [30]. During this process, FAS could add two additional carbon units to malonyl-ACP to produce either 16:0-ACP or 18:0-ACP after six of seven cycles, respectively [18,31]. Furthermore, 18:0-ACP was converted to 18:1-ACP by SAD, then the free C18:1 was released from C18:1-ACP by acyl-ACP thioesterase A (FATA) and the free C16:0 was released from C16:0-ACP by FATB. The free FAs were esterified by ACSL to form an acyl-CoA pool [32,33] which could combine with glycerol-3-phosphate (G3P) to generate triacylglycerol (TAG). KASIII catalyzed the first condensation reaction of FA biosynthesis, while KAR was the first enzyme participating in each cycle of chain elongation [34]. In summary, clarifying the function of key genes may provide scientific support for improving the oil content.

Combined with the information we mentioned above, it is clear that *S. tonkinensis* is an excellent and exploitable tree species with high oil content. Accordingly, understanding the mechanism of the FAs’ biosynthesis is essential for the future improvement of *S. tonkinensis* oil quantity and quality. This study investigated the dynamic changes in FAs’ concentration and composition under EBL5 treatment at four representative time points and attempted to reveal the key genes and underlying mechanism of FA biosynthesis. Our study highlighted that C16:0, C18:1, C18:2, and C18:3 were four major compositions of *S. tonkinensis* during seed development, and the important genes associated with FA biosynthesis presented similar expression patterns. *FATA* was essential for FA metabolism and was affected by EBL unidirectionally, which provided hints at the putative targets of EBL for controlling the lipid productivity [35]. However, *FATA* was not the main gene regulating FA biosynthesis in our study, and we would deeply explore the broader biological functions of *FATA* in a future study. *FATB* not only played an important role in the saturation and carbon chain length of plant FAs but also directly determined the type and quantity of FAs in plant cell lipids [36,37]. It was responsible for the release of free SFA and was expressed lowly at the last two sampling times (100 DAF and 130 DAF), consistent with the results that the concentration of tetradecanoic acid (C14:0), C16:0, and C18:0 was low (Appendix A). EBL5 enhanced the expression of *FATB* (Figure 9), which could be an effective way to improve SFA concentration in *S. tonkinensis* seeds. From 50 DAF to 70 DAF, *KAS III* and *KAR* showed a drastic increase in expression levels, and then a relatively slow decrease between 70 DAF and 130 DAF (Figure 9). Simultaneously, the activity of the FAS encoded by these two genes also displayed the same change trend. It was implied that in the early stage of seed development, the high expression of *KASIII* and *KAR* could enhance FAS activity, subsequently contributing to the biosynthesis of FAs. The expression levels of the two genes under EBL5 treatment were higher than CK. It was concluded that EBL5 treatment had a great effect on the accumulation of FAs in *S. tonkinensis* seeds, both at the physiological and molecular levels. The relatively high expression of these genes at 50 DAF and 70 DAF was synchronized with the considerable increase in oil. This phenomenon partly explained why the FA concentration accumulated rapidly at the early stages, which was identical to the research of *Camellia oleifera* [38]. *FAD2* could catalyze the conversion of C18:1 to C18:2 and was imperative for the desaturation of the UFA biosynthesis [39,40]. The maximum expression of *FAD2* occurred in the middle and late stages of seed development, and the values in CK and EBL5 were 29.2% and 34.4%, respectively. Our results emphasized that the higher the expression of *FAD2*, the higher the concentration of C18:2 (Appendix A, Figure 9). The positive correlation between *FAD2* and C18:2 was confirmed in our experiment. Additionally, the high expression of *FAD2* led to more C18:2 accumulation in *Artemisia sphaerocephala* and *Gossypium hirsutum* seeds [30,41]. ACCase in plants included heterogeneous and homogenous forms, and heterogeneous ACCase was an indispensable speed-limiting enzyme in FA biosynthesis, consisting of four subunits: BC, BCCP, α-CT, and β-CT [42,43]. BCCP was encoded by *accB* and regulated lipid metabolism and EBL response in *A. thaliana* [35], while α-CT was encoded by *accA*. In this experiment, both *accB* and *accA* in seeds from EBL5-treated sample trees had a higher expression at 70 DAF, inducing more FA accumulation in this period.

To sum up, exogenous EBL treatment is an efficient and viable method to promote FA accumulation via increasing the expression of FA biosynthesis-related genes. Identifying the expression levels of various genes at different developmental stages of *S. tonkinensis* seeds could lay the foundation for future research on the molecular mechanism of FA biosynthesis and molecular improvement of FA compositions.

### 3.3. Transcription Factors Involved in Fatty Acid Biosynthesis

Transcription factors (TFs) operate in gene regulatory networks by controlling the expression of crucial genes coding for some enzymes involved in FA biosynthesis in plant seeds [44]. *GRF5*, *WRI1*, and *FUS3* played a role in the oil biosynthesis regulatory network in *Brassica rape* seeds [45], however, *PBS* and *RAP* were primary TFs in oil biosynthesis in *Persea americana* seeds [46]. As described in Figure 11, *MYB* (213), *C2H2* (141), *AP2/ERF* (130), *bHLH* (108), *GRAS* (106), *bZIP* (99), *C2C2* (94), and *WRKY* (75) dominated in the FA biosynthesis of *S. tonkinensis* seeds. Thus, the basilic TFs regulating oil biosynthesis varied by species [30]. *bZIP* could bind to the *FAD3* promoter and activate its expression to synthesize α-linolenic polyunsaturated FAs in *A. thaliana* seeds [47]. *MYB96* could promote oil accumulation by activating the expression of *DGAT1* and *PDAT1*, so *myb96* mutants accumulated less oil [48]. *AP2* was considered a negative regulator in oil accumulation, as was *WRKY6* [49,50]. Here, *MYB*, *AP2*, *bZIP*, and *WRKY* were major TFs in *S. tonkinensis* seeds. The overexpression of positive regulators through overexpression constructs using constitutive promoters helped increase FA concentration in seeds [44]. How these TFs cause a difference in FA biosynthesis in *S. tonkinensis* seeds remains to be studied, which includes causing an alteration in the FA composition, changing the FA concentration, or different TFs belonging to the same gene family regulating the FA biosynthesis positively or negatively.

## 4. Materials and Methods

### 4.1. Site Condition and Plant Materials

The *S. tonkinensis* plants (from Jishui, Jiangxi Province) were purchased by Jiangsu Guoxing Co., Ltd., in 2011, and planted in the Styracaceae Germplasm Repository, Luhe District, Nanjing, China (32°54′ N, 118°50′ E). The average temperature, hours of sunshine, and rainfall per year were 15.3 °C, 2200 h, and 970 mm, respectively. The experimental site is hilly land, and the soil fertility is suitable for plant growth. The trees grew under natural environmental conditions and no fertilizer was applied. *S. tonkinensis* starts to flower in late May and quickly enters the blooming stage.

### 4.2. Experimental Design and Treatment

Our past experiments manipulated four concentrations of EBL (1, 5, 10, and 20 μmol/L) to spray the whole trees in order to study their effect on FA concentration in seeds. Based on the experimental results, we found that the FA concentration in seeds from EBL-treated trees peaked at 70 DAF, and 5 μmol/L EBL (marked as EBL5), 10 μmol/L EBL (marked as EBL10) as well as 20 μmol/L EBL (marked as EBL20) boosted the FA accumulation, of which EBL5 resulted in the highest FA concentration and was the most significant treatment. Therefore, we selected samples treated by EBL5 for transcriptome and GC-MS analysis. On 30 June 2020 (40 DAF), 12 trees with similar height, growth, and good condition were selected and tagged. In total, we designed two treatments, including CK and EBL5, so each treatment contained six trees.

EBL (purchased from Shanghai Yuanye Biological Technology Co., Ltd., Jinshan, Shanghai, China) was dissolved in a small volume of alcohol and then diluted with distilled water into 5 μmol/L. Approximately 1800 mL of the EBL was sprayed on each sampled tree (spray on the whole tree). The CK plants were treated with an equal amount of distilled water. Exogenous treatment was applied every ten days from 45 DAF to 135 DAF and was always performed between 6:00 to 8:00 A.M.

### 4.3. Sample Collection

On the basis of our previous research on the seed development of *S. tonkinensis* [17,18], we selected samples from four representative time points, including 50 (the previous stage before the seed dry matter rapid increase), 70 (during the aforementioned steep rise in nutrient concentration), 100 (the stage with decreasing oil concentration and increasing starch concentration) and 130 DAF (the final maturation stage).

About 10 seeds were randomly collected from each individual tree at each sampling time, and the seeds from one treatment were mixed. These samples were placed on an insulated box with dry ice for transport to the laboratory. Therefore, 60 seeds (divided into three replicates) were used for the FA concentration and composition analysis. Besides, 5 seeds from each tree were sampled, and every two trees were regarded as one biological replicate so that three biological replicates in each treatment with 10 seeds in each replicate were used for transcriptome analysis. The samples for transcriptome analysis were immediately frozen in liquid nitrogen and stored at −80 °C until use.

### 4.4. Fatty Acid Extraction and GC-MS Analysis

Forty-five seeds per treatment were dried at 65 °C for 72 h. Then, the seeds were crushed and transferred into filter paper, which was dried and weighed (M0). The seeds and filter paper were weighed (M1) after drying for 30 min. Petroleum ether (30–60 °C) was used as the extraction solution and the Soxhlet apparatus was adopted to extract the FAs. The extraction process lasted for 24 h, and the water temperature was controlled at 65 °C. After extraction, the filter paper covering the seeds was dried at 65 °C for 30 min (M2). Calculate the FA concentration according to the formula below:$$\mathrm{FA}\;\mathrm{concentration}\;(\%)=\frac{M1-M2}{M1-M0}\times 100\%$$

The solid seeds (0.2 g) were accurately weighed, and the FA compositions were extracted using 0.5 mL of methanol and 0.5 mL of chloroform with a steel ball. The mixture was homogenized at 50 Hz for 3 min. After being set for 15 min, the mixture was centrifuged at 13,000 rpm for 10 min. Briefly, 1 mL of dichloromethane was added to the supernatant and the above steps were repeated. Add 1 mL of sodium hydroxide-methanol solution to the mixed solution, shake for 30 secs, and place the mixture in a 60 °C water bath for 30 min. Then, 1 mL of n-hexane and 1 mL of sodium hydroxide were added after the water bath was cooled. The mixture was centrifuged at 13,000 rpm for 10 min. Transfer the upper solution to a 1.5 mL centrifuge tube, add 2 mL of n-hexane and shake for 30 sec, then centrifuge the mixture at 13,000 rpm for 5 min at 4 °C. The supernatant was transferred into GC vials. The GC-MS analysis was performed on an Agilent 8890B-5977B GC-MS system (Agilent Technologies Inc., Santa Clara, CA, USA) coupled with an Agilent DB-FastFAME column (20 m, 0.18 mm, 0.2 μm, Agilent J&W Scientific, Folsom, CA, USA). Referring to our previous studies [51,52], we performed minor modifications to the measurement process.

The GC conditions were as follows: Helium (purity > 99.999%) was used as carrier gas at a flow of 1 mL/min. The injection volume of the sample was 1 μL and was introduced in splitting mode (50:1) with the injector temperature of 250 °C. The GC column temperature was programmed to hold at 80 °C for 0.5 min and rose to 180 °C at a rate of 70 °C per minute, then increased to 220 °C at a rate of 4 °C per minute, and finally held at a temperature of 240 °C for 2 min. The MS conditions were as follows: An electro ionization system was used with 70 eV of ionization energy. The ion source temperature was 230 °C, the quadrupole temperature was 150 °C, and the transmission line temperature was 240 °C. The data acquisition was conducted on the selective ion scan mode. The compounds were identified and quantified by the software of Masshunter (v10.0.707.0, Agilent Technologies Inc., Santa Clara, CA, USA) with manual inspection. A linear regression standard curve was created with the mass spectrum peak area of the analyte as the ordinate and the concentration of the analyte as the abscissa. The mass spectrum peak area of the analyte was substituted into the linear equation in order to calculate the sample’s concentration.

### 4.5. Enzyme Activity Assay

Briefly, 0.5 g of seeds were weighed to measure the FAS activity with an assay kit (Suzhou Comin Biotechnology Co., Ltd., Suzhou, Jiangsu, China). As described by [53], the samples were homogenized in buffer containing 1 mmol/L EDTA-Na_2_, 2 mmol/L AsA and 20 mmol/L Tris-HCl (pH 7.5). The reactions were performed in a total volume of 1000 μL (100 μL of supernatant, 20 μL of acetyl-CoA, 20 μL of malonyl-CoA, 820 μL of 30 °C PBS (pH 7.0), and 40 μL of NADPH). Decreases in NADPH absorbance at 340 nm were detected with a spectrophotometer, and the activity of FAs was calculated by the method of [54].

### 4.6. RNA Extraction

The Total RNA was extracted from the kernel tissue using Plant RNA Purification Reagent (Invitrogen, Carlsbard, CA, USA) according to the manufacturer’s instructions, and the genomic DNA was removed using DNase I (TaKara). Then, the integrity and purity of the total RNA quality were determined by a 2100 Bioanalyser (Agilent Technologies, Inc., Santa Clara CA, USA) and quantified using the ND-2000 (NanoDrop Thermo Scientific, Wilmington, DE, USA). Only high-quality RNA samples (OD260/280 = 1.8~2.2, OD260/230 ≥ 2.0, RIN ≥ 8.0, 28S:18S ≥ 1.0, >1 μg) ere used to construct sequencing library.

### 4.7. Library Preparation and Illumina Hiseq Xten/NovaSeq 6000 Sequencing

The RNA purification, reverse transcription, library construction, and sequencing were performed at Shanghai Majorbio Bio-pharm Biotechnology Co., Ltd. (Pudong, Shanghai, China) according to the manufacturer’s instructions (Illumina, San Diego, CA, USA). The RNA-seq transcriptome libraries were prepared using an Illumina TruSeqTM RNA sample preparation Kit (San Diego, CA, USA). Poly(A) mRNA was purified from total RNA using oligo-dT-attached magnetic beads and then fragmented by fragmentation buffer. Taking these short fragments as templates, double-stranded cDNA was synthesized using a SuperScript double-stranded cDNA synthesis kit (Invitrogen, Carlsbard, CA, USA) with random hexamer primers (Illumina). Then, the synthesized cDNA was subjected to end-repair, phosphorylation, and ‘A’ base addition according to Illumina’s library construction protocol. Libraries were size selected for the cDNA target fragments of 200–300 bp on 2% Low Range Ultra Agarose followed by PCR amplified using Phusion DNA polymerase (New England Biolabs, Boston, MA, USA) for 15 PCR cycles. After being quantified by TBS380, the RNAseq libraries were sequenced in a single lane on an Illumina NovaSeq 6000 sequencer (Illumina, San Diego, CA, USA) for 2 × 150 bp paired-end reads.

### 4.8. De Novo Assembly and Annotation

The raw paired-end reads were trimmed and quality controlled by SeqPrep (https://github.com/jstjohn/SeqPrep, accessed on 24 April 2020) and Sickle (https://github.com/najoshi/sickle, accessed on 24 April 2020) with the default parameters. Then, clean data from the samples were used for de novo assembly with Trinity (http://trinityrnaseq.sourceforge.net/, accessed on 28 April 2020) [55]. All the assembled transcripts were searched against the NR, COG, and KEGG databases using BLAST to identify the proteins that had the highest sequence similarity with the given transcripts to retrieve their function annotations, and a typical cut-off of E-values less than 1.0 × 10^−5^ was set. The BLAST2GO (http://www.blast2go.com/b2ghome, accessed on 4 May 2020) [56] program was used to obtain GO annotations of unique assembled transcripts for describing biological processes, molecular functions, and cellular components. Metabolic pathway analysis was performed using the KEGG (http://www.genome.jp/kegg/, accessed on 4 May 2020) [57,58].

### 4.9. Differential Expression Analysis and Functional Enrichment

To identify DEGs between different samples, each sample was compared to each gene to obtain the number of reads, and the transcripts per million reads (TPM) conversion was conducted to obtain the expression level of the transcript. The TPM was calculated by standardizing gene length first, then sequencing depth, and finally by obtaining a read count matrix. RNA-Seq by expectation-maximization (RSEM) (http://deweylab.biostat.wisc.edu/rsem/, accessed on 7 July 2020) was used to quantify gene abundances [59]. Essentially, differential expression analysis was performed using the R package with DESeq2, when |log2FC| > 1 and Q value ≤ 0.05, it was considered to be a significantly differently expressed gene [60]. In addition, functional enrichment analyses including GO and KEGG were performed to identify which DEGs were significantly enriched in GO terms and metabolic pathways at a Bonferroni-corrected *p*-value ≤ 0.05 compared with the whole-transcriptome background. The GO functional enrichment was carried out by Goatools (https://github.com/tanghaibao/Goatools, accessed on 24 May 2020), and Fisher’s test was used. The GO terms with *p* ≤ 0.05 were selected as significantly enriched GO terms. When the significant GO term screened by the classic Fisher is less than 0.05 (threshold value), the false discovery rate (FDR) value obtained after the *p*-value correction could be utilized for screening the GO term, and the threshold value was also 0.05. KEGG pathway analysis was performed by KOBAS (https://kobas.cbi.pku.edu.cn/home.do, accessed on 24 May 2020) [61]. Similar to the GO enrichment analysis, Fisher’s test was adopted and the Benjamini-Hochberg (BH) method was used for the pathway enrichment analysis. In general, when the *p*-value corrected by BH is less than 0.05, the KEGG pathways satisfying the condition were defined as significantly enriched KEGG pathways.

### 4.10. qRT-PCR Validation

We selected ten genes related to oil accumulation for validation using qRT-PCR. The amplification primers were designed by Primer Premier 5.0 software (Premier Biosoft International, Palo Alto, CA, USA). All reactions were carried out with a StepOne Real-Time PCR System (Applied Biosystems, Foster City, CA, USA) and SYBR Green Premix Pro Taq HS qPCR Kits (AG11701, Accurate Biotechnology, Hunan, Co., Ltd., Changsha, Hunan, China). The relative gene expression was calculated by the 2^−ΔΔCt^ method with 18S ribosomal RNA as an internal control. All primers used in this study were listed in Appendix A.

### 4.11. Statistics Analysis

The values were expressed as mean ± SD for three replicates. Excel (Office 2019 Pro Plus, Microsoft Corporation, Redmond, WA, USA) was used to process the data. One-way analysis of variance (ANOVA) was performed using SPSS 26.0 (IBM, Armonk, NY, USA) followed by Duncan’s multiple range test. *p*-values less than 0.05 were considered to indicate significance within groups.

## 5. Conclusions

The current study illustrated that EBL5 increased the FA concentration and FA content per seed during *S. tonkinensis* seed development, and the FA compositions remained relatively static, among which the UFAs’ concentration was much higher than that of SFAs. Hence, the exogenous application of EBL5 could be considered a feasible approach to boost the yield and quality of the seed oil of *S. tonkinensis*. The high expression of FA biosynthesis-related genes (*KAR*, *KASIII*, and *accA*) peaked at 70 DAF, consistent with FA biosynthesis-related enzyme and FA concentration. *FAD2* was expressed highly in the middle and late stages of seed development and was helpful for C18:2 accumulation, which was a crucial UFA in *S. tonkinensis* seeds. In addition, *FATB* was closely relevant to the SFA concentration. EBL5 promoted the expression of genes that encoded the enzyme related to FA biosynthesis, thereby increasing the FA concentration. *MYB*, *AP2*, *bZIP*, and *WRKY* were major TFs and dominated the FA biosynthesis of *S. tonkinensis* seeds. Our results could provide guidance to understand the molecular mechanism of FA biosynthesis regulated by EBL in seeds and ample genetic resources for the molecular research of a potential biodiesel species.

## Figures and Tables

**Figure 1 ijms-23-08897-f001:**
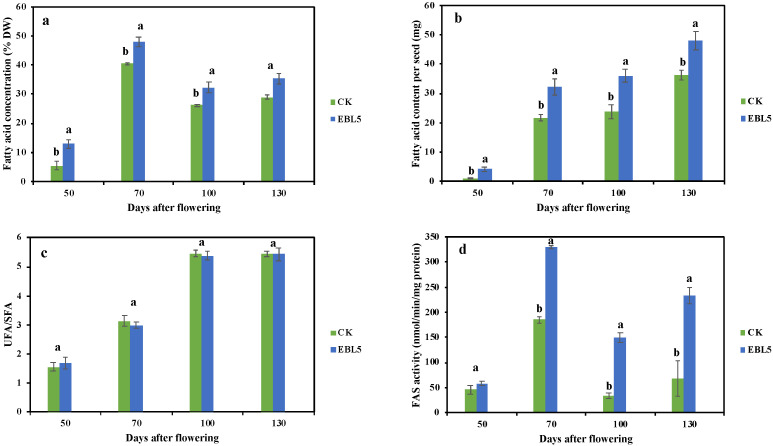
Effects of EBL5 on FA concentration, UFA/SFA ratio, and FA biosynthesis-related enzyme activity in *S. tonkinensis* seeds during seed development. (**a**) FA concentration in *S. tonkinensis* seeds at 50, 70, 100, and 130 DAF; (**b**) Fatty acid content in single *S. tonkinensis* seeds at 50, 70, 100, and 130 DAF; (**c**) UFA/SFA ratio in *S. tonkinensis* seeds at 50, 70, 100, and 130 DAF; (**d**) FA biosynthesis-related enzyme activity in *S. tonkinensis* seeds at 50, 70, 100, and 130 DAF. Data points showed mean ± SD; Different lowercase letters indicated significant differences (*p* < 0.05).

**Figure 2 ijms-23-08897-f002:**
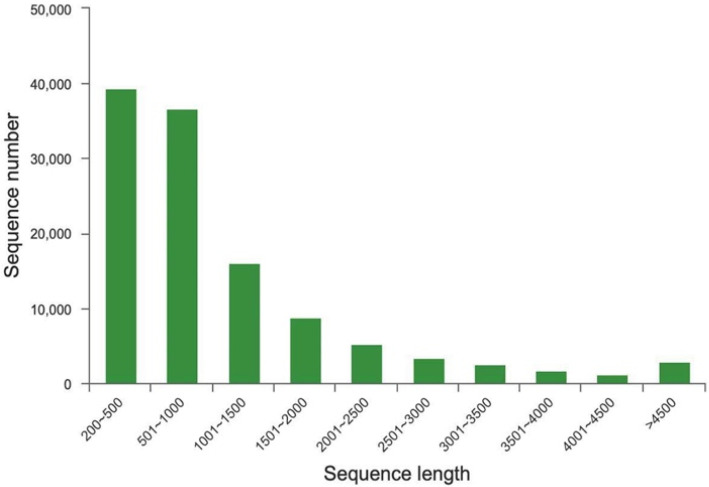
Sequence length distribution.

**Figure 3 ijms-23-08897-f003:**
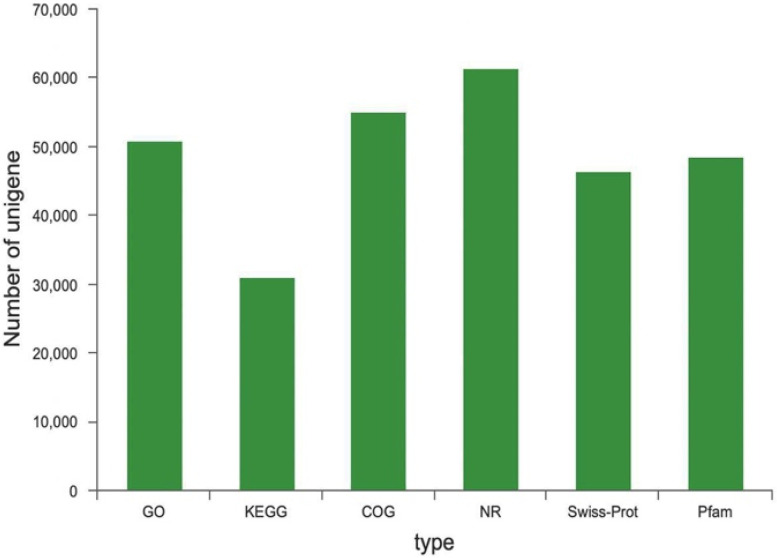
Functional annotation of unigenes.

**Figure 4 ijms-23-08897-f004:**
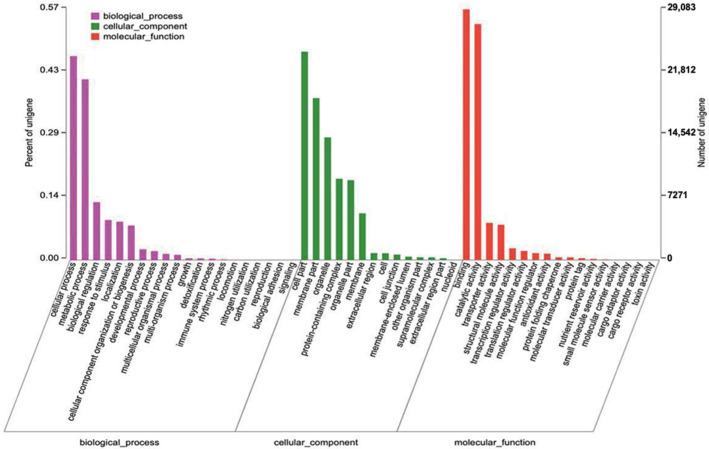
GO classification of unigenes.

**Figure 5 ijms-23-08897-f005:**
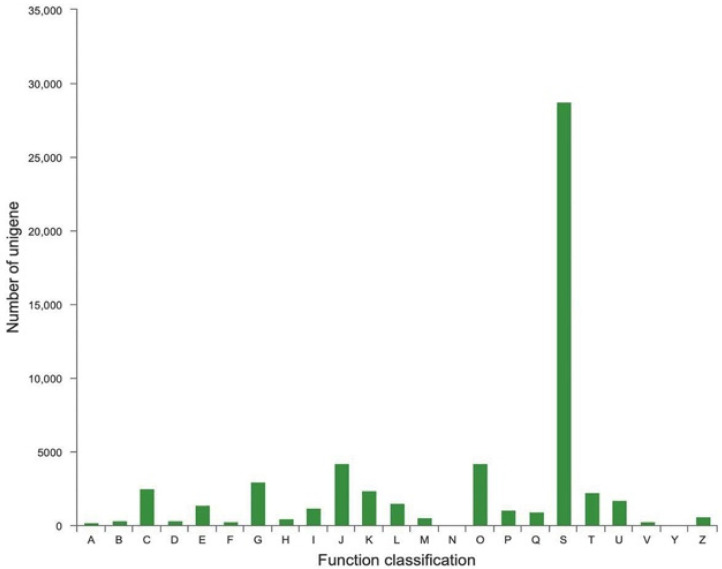
COG classification of unigenes. A: RNA processing and modification; B: Chromatin structure and dynamics; C: Energy production and conversion; D: Cell cycle control, cell division, chromosome partitioning; E: Amino acid transport and metabolism; F: Nucleotide transport and metabolism; G: Carbohydrate transport and metabolism; H: Coenzyme transport and metabolism; I: Lipid transport and metabolism; J: Translation, ribosomal structure, and biogenesis; K: Transcription; L: Replication, recombination, and repair; M: Cell wall/membrane/envelope biogenesis; N: Cell motility; O: Posttranslational modification, protein turnover, chaperones; P: Inorganic ion transport and metabolism; Q: Secondary metabolites biosynthesis, transport, and catabolism; S: Function unknown; T: Signal transduction mechanisms; U: Intracellular trafficking, secretion, and vesicular transport; V: Defense mechanisms; Y: Nuclear structure; Z: Cytoskeleton.

**Figure 6 ijms-23-08897-f006:**
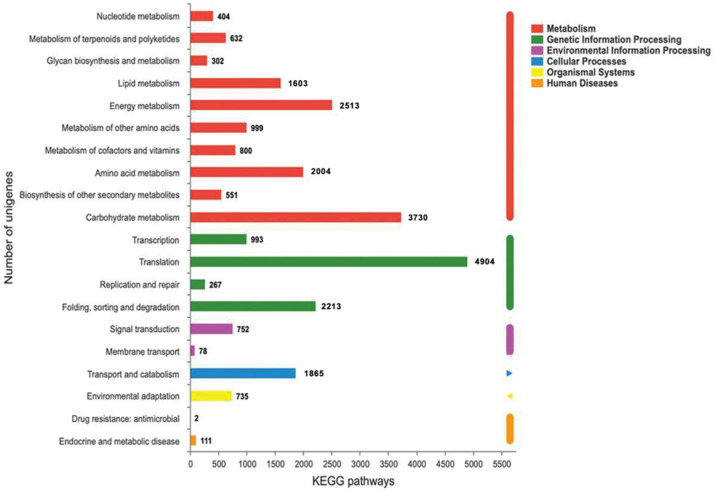
KEGG classification and pathway assignment of unigenes.

**Figure 7 ijms-23-08897-f007:**
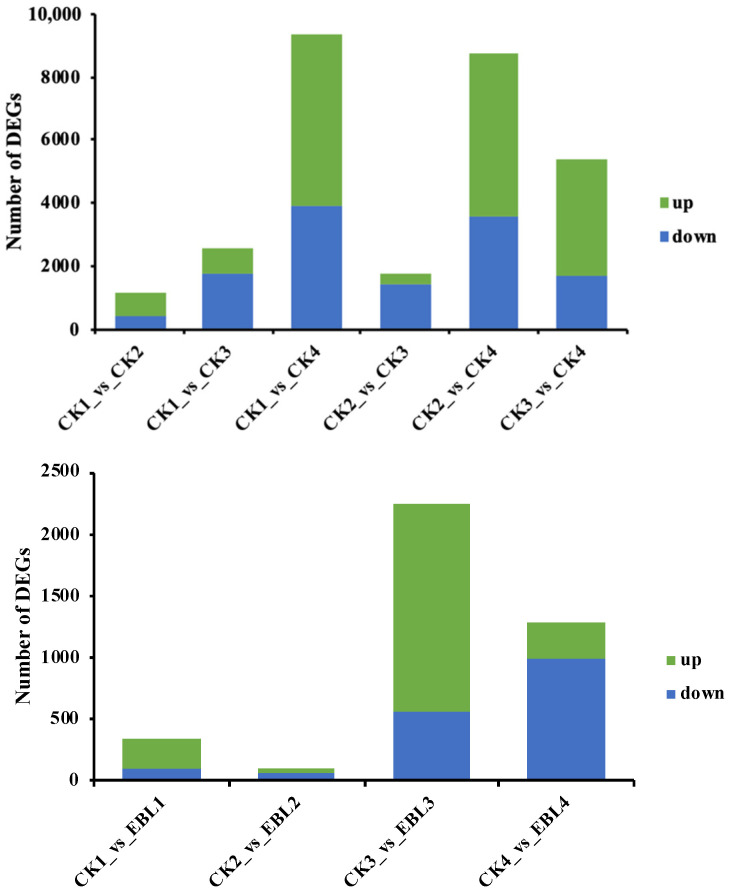
Number of upregulated and downregulated unigenes under EBL5 at four sampling times.

**Figure 8 ijms-23-08897-f008:**
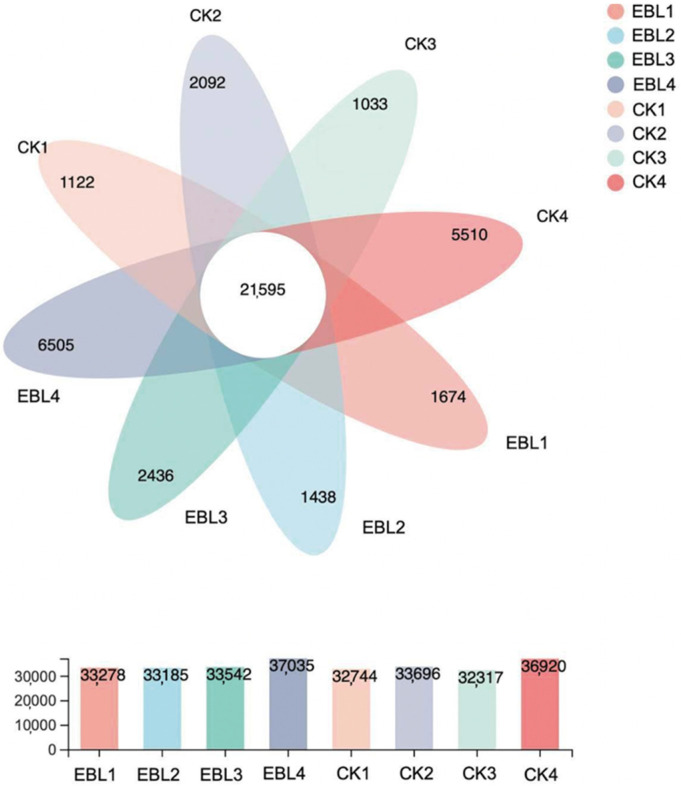
Number and distribution of co-expression and specific expression genes.

**Figure 9 ijms-23-08897-f009:**
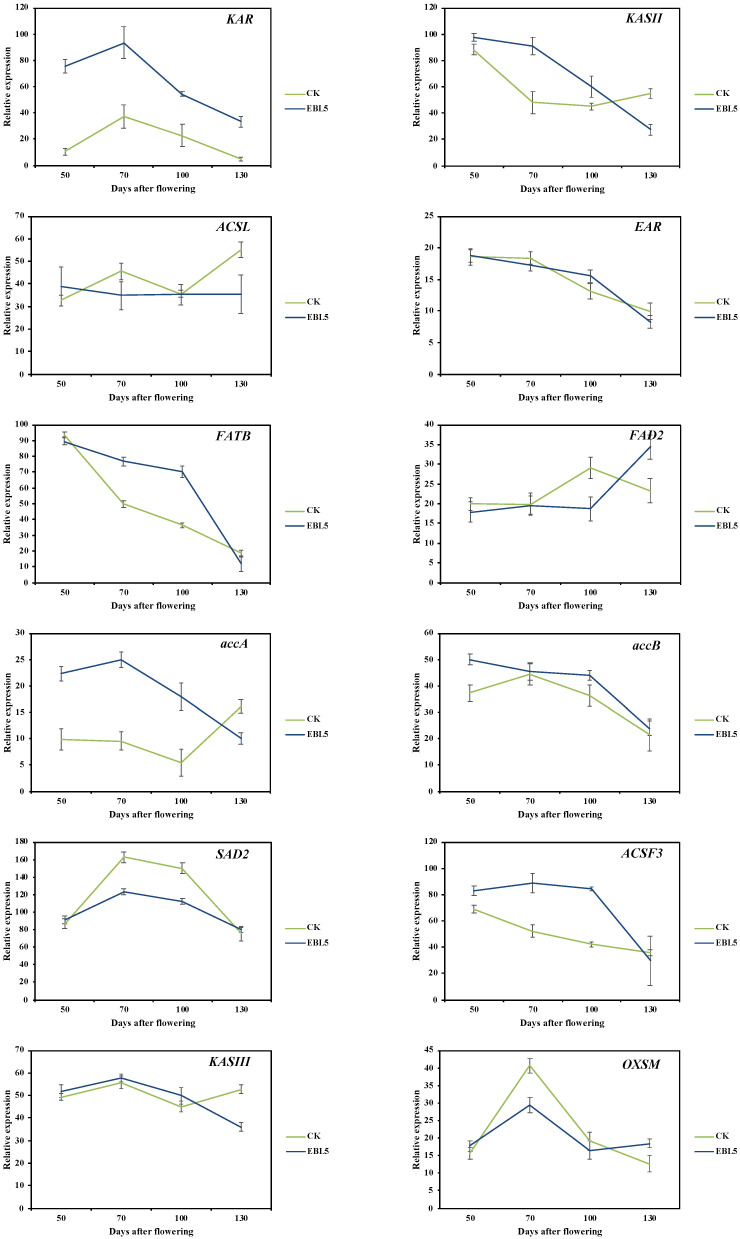
Relative expression levels of FA biosynthesis-related genes at 50, 70, 100, and 130 DAF. Data were shown as mean ± SD of three biological replicates; gene expression levels at 50 DAF were used as reference.

**Figure 10 ijms-23-08897-f010:**
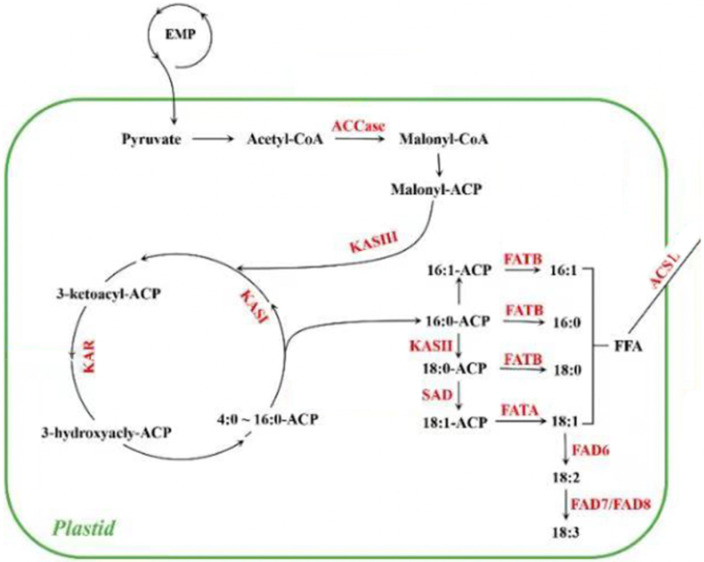
FA biosynthesis pathway.

**Figure 11 ijms-23-08897-f011:**
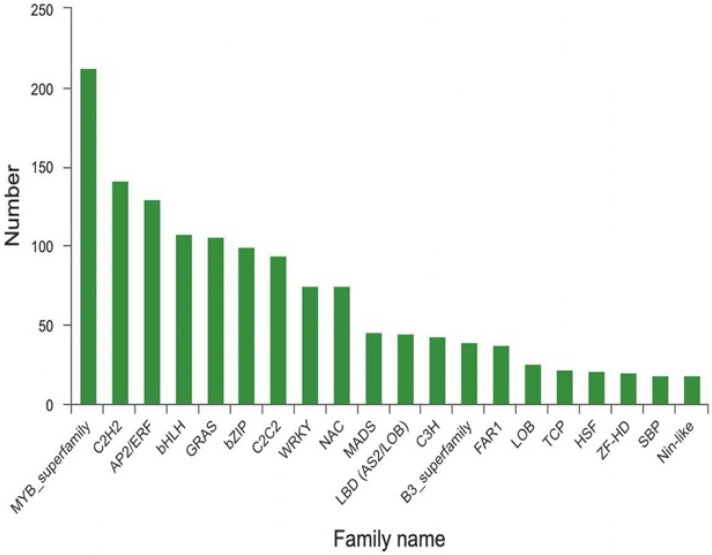
Transcription factors involved in FA biosynthesis. TFs were compared with PlantTFDB 4.0; *MYB*, *C2H2*, *AP2/ERF*, *bHLH*, *GRAS*, *bZIP*, *C2C2*, and *WRKY* were screened as the main TFs involved in the biosynthesis of FA in the seeds of *S. tonkinensis*, and they were sequenced by the number of unigenes in a TF family.

**Table 1 ijms-23-08897-t001:** Statistics of raw data and clean data.

Sample	DAF	Library ID	Raw Data		Clean Data		Clean Ratio (%)
			Reads	Bases	Reads	Bases	
CK	50	ST1	52,576,118	7,938,993,818	52,069,462	7,661,446,106	99.04
		ST2	58,412,370	8,820,267,870	58,030,022	8,551,294,813	99.35
		ST3	49,291,606	7,443,032,506	48,938,812	7,185,882,099	99.28
	70	ST4	51,841,414	7,828,053,514	51,474,434	7,555,435,732	99.29
		ST5	51,238,464	7,737,008,064	50,877,488	7,488,832,185	99.30
		ST6	44,519,776	6,722,486,176	44,152,016	6,490,325,526	99.17
	100	ST7	52,769,788	7,968,237,988	52,359,108	7,645,287,938	99.22
		ST8	45,504,386	6,871,162,286	45,000,182	6,644,260,279	98.89
		ST9	52,755,456	7,966,073,856	52,287,670	7,623,692,244	99.11
	130	ST10	54,480,322	8,226,528,622	54,005,880	7,888,438,631	99.13
		ST11	50,006,582	7,550,993,882	49,610,496	7,278,763,737	99.21
		ST12	58,199,588	8,788,137,788	57,780,438	8,505,039,657	99.28
EBL5	50	ST13	48,743,066	7,360,202,966	48,351,776	7,148,014,849	99.20
		ST14	53,107,548	8,019,239,748	52,747,854	7,781,457,002	99.32
		ST15	52,110,960	7,868,754,960	51,625,538	7,579,374,419	99.07
	70	ST16	48,385,070	7,306,145,570	48,059,442	7,107,199,135	99.33
		ST17	48,352,048	7,301,159,248	47,947,800	7,060,381,020	99.16
		ST18	48,578,174	7,335,304,274	48,188,350	7,109,430,178	99.20
	100	ST19	47,260,856	7,136,389,256	46,768,994	6,894,657,098	98.96
		ST20	40,949,022	6,183,302,322	40,313,670	5,858,080,882	98.45
		ST21	47,964,600	7,242,654,600	47,474,716	6,980,363,633	98.98
	130	ST22	56,293,326	8,500,292,226	55,793,860	8,189,033,282	99.11
		ST23	50,022,084	7,553,334,684	49,528,614	7,341,248,924	99.01
		ST24	48,076,766	7,259,591,666	47,675,840	7,001,365,324	99.16

**Table 2 ijms-23-08897-t002:** Summary of do novo transcriptome assembly for *S. tonkinensis* seeds.

Type	Unigenes
Total number	117,904
Total base	132,077,283
Largest length (bp)	16,018
Smallest length (bp)	201
Average length (bp)	1120
N50 length (bp)	1705
Fragment mapped percent (%)	60.66
GC content (%)	42.21

## Data Availability

Raw sequence data were deposited in the NCBI Short Read Archive database under accession number PRJNA791699 (https://www.ncbi.nlm.nih.gov/bioproject/PRJNA791699, accessed on 7 August 2022).

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
