# Peer review of "24-Epibrassinolide Promotes Fatty Acid Accumulation and the Expression of Related Genes in Styrax tonkinensis Seeds"

_ijms, 2022, doi:10.3390/ijms23168897_

Round 1

Reviewer 1 Report

In this paper, the authors found 24-epibrassinolide treatment promotes fatty acid accumulation in Styrax tonkinensis seeds and performed RNAseq analysis in 24-epibrassinolide-treated Styrax tonkinensis. In other tree species, some researchers have found the accumulation of fatty acid by 24-epibrassinolide treatment. In addition, although the authors analyzed the functional annotation, GO classification, COG classification and KEGG classification of contigs obtained by de novo assembly in Fig. 3 to Fig. 7, similar analyses have been already reported by the authors (Wu et al., 2020). While the most important points are the analysis of DEGs by 14-epibrassinolide treatment in this study, few analyses are described in this paper. Thus, the reviewer thinks that this paper falls short of IJMS’s standard for novelty.

Also, enough information to understand Figure 10 is not provided in Figure legend. What do the error bars indicate? Is this fugure the results of RNAseq analysis or qRT-PCR analysis?

Reviewer 2 Report

In this study, authors conducted RNA sequencing and GC-MS with Styrax tonkinesis seeds revealing that 24-Epibrassinolide promotes fatty acid accumulation. In general, the manuscript was nicely written although there were some mistakes. Analyses associated with differentially expressed genes (DEGs) were not well described. DEG analysis should be reanalyzed with read counts instead of RPKM or TPM values. My comments are as follows.

L19 among which palmitic acid, -> among them, palmitic acid,

L20 1,120 bp    1,205 unigenes

L22-23 Twelve impportant genes related to FA biosynthesis were identified and their expression was confirmed by qRT-PCR. Please spell out qRT-PCR

L26 EBL5 induced expression of the FA biosynthesis-related genes.

L27 The concentration of FA was increased after EBL5 application.

L59 a member of the family Styracaceae

L85 at four time points -> Please specify four time points in detail.

Figure 1 can be modified as 2 X 2 dimensional array to reduce unnecessary space.

In the Table 1, please provide accession numbers of each library. I found that authors used three biological replicates; however, authors did not mention about that. Please described it in the manuscript.

L143 1,120 bp

L146 4,500 bp        1,001-4,500 bp

The title of Table 2 should be changed as “Summary of de novo transcriptome assembly for Styrax tonkinesis seeds.  De novo should be in italics.

How about to combine Figure 2 and 3 making a figure panel?

Figure 4 should be magnified with the high-quality image.

L156 BLASTX   Authors have done BLASTX search.

L181 4,232 and 4,199 unigenes   1,205 unigenes

L183 candidates -> candidate genes

Figures 5 and 6 were too small to see. Please provide high quality images.

L206-244 I am not sure whether authors conducted DEG analysis correctly. Otherwise this paragraph should be rewritten.

L235 Here we performed Venn analysis to obtain the co-expression and specific expression -> An awkward expression.

L249 We subjected DEGs identified in the current study to KEGG pathway enrichment -> An awkward expression.

How did you obtain the results for Figure 9? Please explain it in the figure legend and results.

Please describe about Figure 10 in detail in the figure legend.

L425-430 Remove space and make a sentence.

L492-507 Please explain about libraries in detail. How many libraries were generated for a condition?

L511 to do de novo assembly -> for de novo assembly      de novo in italics

L520-530 For DEG analysis, authors should use the number of mapped reads (read count). Authors described that they used the DEGs. What is DEGs? Please specify the name of program. To identify DEGs, DESeq or edgeR programs are used. For cutoff, log2FC and adjusted pvalues should be used. Please reanalyse DEGs using read count and revised the associated results.

I found that there were several awkward expressions in the manuscript. The manuscript should be edited by the English editing service.

Round 2

Reviewer 1 Report

As the authors claim, the important point of this study is that brassinolide treatment increased fatty acid content. The authors may have performed RNAseq to analyze the mechanism. If so, it is very important that analysis of the DEGs that respond to brassinolide treatment, e.g., which pathway genes show significant expression variation, whether the expression of brassinolide-responsive genes known in other plants is the same or different, and how these results lead to changes in fatty acid content. However, even in the revised version, the authors have not provided more explanations in figures and tables for the analysis of DEGs. The phenomena found by the authors are very interesting, but they are not up to IJMS standards, and the paper does not deserve to be published if there is no analysis of the mechanism.

Reviewer 2 Report

Authors properly revised their manuscript according to reviewers' comments.

I suggest the manuscript for publication as it is. 

Round 3

Reviewer 1 Report

It is better to describe the expression levels of all genes in Supplementary Tables so that the readers can use the data presented in this manuscript.   
